# Understanding Partial Multi-label Learning via Mutual Information

**Xiuwen Gong** [†,‡], **Dong Yuan**[†], **Wei Bao** [†]

[†] Faculty of Engineering, The University of Sydney

[‡] Hunan Huishiwei Intelligent Technology Co., Ltd.

{xiuwen.gong, dong.yuan, wei.bao}@sydney.edu.au

## Abstract

To deal with ambiguities in partial multi-label learning (PML), state-of-the-art methods perform disambiguation by identifying ground-truth labels directly. However, there is an essential question:"*Can the ground-truth labels be identified precisely?*". If yes, "*How can the ground-truth labels be found?*". This paper provides affirmative answers to these questions. Instead of adopting hand-made heuristic strategy, we propose a novel **M**utual **I**nformation **L**abel **I**dentification for **P**artial **M**ulti-Label **L**earning (**MILI-PML**), which is derived from a clear probabilistic formulation and could be easily interpreted theoretically from the mutual information perspective, as well as naturally incorporates the feature/label relevancy into consideration. Extensive experiments on synthetic and real-world datasets clearly demonstrate the superiorities of the proposed MILI-PML.

## 1 Introduction

Partial multi-label learning (PML) [1, 2] is a weakly supervised learning problem, where each instance is associated with a set of candidate labels, but only a part of them are the ground-truth labels while others are false positive labels.

In recent years, many real-world applications are arising due to the growing demand for identifying ground-truth labels from partially labeled data, which are easier and less costly to obtain. For example, in crowdsourcing image annotation, a web image might be annotated online by a potential unreliable annotator with many specific labels, but only some of them are accurate.

PML aims to train a classifier from partially labeled data so as to predict the ground-truth labels for an unseen instance automatically. The main challenge is how to deal with the ambiguities caused by false positive labels in candidate label set. One straightforward way is to simply treat all candidate labels equally as the ground-truth labels, and then solve the PML problem by standard multi-label classification methods [3, 4]. However, these methods can be easily misled by the noisy false-positive labels in the candidate set, and fail to generalize well in testing. To deal with the problem, the state-of-the-art PML research attempts to identify the ground-truth labels directly from the candidate label set, which becomes a popular and effective disambiguation strategy.

However, there are two fundamental issues, "*Can the ground-truth labels be identified precisely?*". If yes, "*How can the ground-truth labels be found?*". Moreover, the state-of-the-art PML methods are constructed by hand-designed heuristic modeling under considerations like feature/label correlations with no theoretical interpretation.

In this paper, instead of hand-made heuristic modeling, we propose a novel MILI-PML model, which is derived from a clear probabilistic formulation and can be easily interpreted theoretically by information theory. Besides, MILI-PML naturally incorporates the feature/label relevancy considerations rather than that of heuristic modeling.

35th Conference on Neural Information Processing Systems (NeurIPS 2021).

To answer the first question "CAN", we define an $\epsilon$-identifiable score function to evaluate whether a PML method is workable for identifying the ground-truth labels precisely. Moreover, we demonstrate that the derived MILI-PML is $\epsilon$-identifiable under certain assumptions and conditions theoretically.

To answer the second question "HOW", we design an efficient optimization algorithm called MILI Algorithm to optimize the objective function.

Extensive experiments are conducted on six synthetic datasets as well as four real-world datasets. The experimental results demonstrate that the proposed MILI-PML consistently outperforms the state-of-the-art methods in terms of five widely-used multi-label metrics including ranking loss, hamming loss, one error, coverage, and average precision.

## 2 Related Work

Partial multi-label learning (PML) [5, 6] is different from multi-label learning [7, 8, 9] or partial label learning [10, 11, 12, 13]. In PML, each instance is associated with a set of candidate labels, which contains multiple ground-truth labels and the others are false positive labels. The state-of-the-art PML studies attempt to identify the ground-truth labels directly from the candidate label set.

[14] propose PARTICLE to extract credible labels with high confidence via propagation matrix and use the identified labels to train multi-label classifiers. [15] propose DRAMA to get the reliable labels with high confidence by employing the feature manifold, and then use the identified labels to train the multi-label classifier.

Moreover, [1] propose PML-lc and PML-fp to optimize the label ranking confidence matrix in training classifiers which considers the label correlations and the feature prototype respectively. [16] develop fPML to optimize the label confidentce matrix by considering feature and label correlations. [17] propose PML-LRS to get label ranking which utilizes the low-rank and sparse decomposition to train classifiers while considering the feature and label interdependencies. [18] develop MUSER to train classifiers by decreasing the feature noise and label redundancy via mapping with orthogonality constraint and graph Laplacian regularization, which considers the feature correlation and label correlation simultaneously.

However, existing methods have two common issues. First, models are hand-made heuristic with no theoretical interpretation. Second, existing partial multi-label learning (PML) methods assume that the ground-truth labels exist in the candidate label set and identify them directly. We then have the question:"*Can the ground-truth labels be identified precisely?"*. If yes, *"How can the ground-truth labels be found?"*.

## 3 The Model Derivation

In this section, we derive our method, i.e., MILI-PML, from the following probabilistic formulation. Assume $\mathcal{S} = \{(\mathbf{x}^i, \mathbf{y}^i) : i = 1, \ldots, n\}$ are drawn i.i.d. $n$ times from a probability distribution $P$. Each observation is a pairwise example $(\mathbf{x}, \mathbf{y})$, consisting of a $d$-dimensional feature vector $\mathbf{x} = [x_1, \ldots, x_d]^T$, and a $q$-dimensional candidate label vector $\mathbf{y} = [y_1, \ldots, y_q]^T$, drawn from the underlying random variables $\mathbf{X} = [X_1, \ldots, X_d]^T$ and $\mathbf{Y} = [Y_1, \ldots, Y_q]^T$.

We adopt a $q$-dimensional binary vector $v$ : a 1 indicating the label is selected, a 0 indicating it is discarded. Notation $\mathbf{y}_v$ indicates the vector of selected labels, that is, the full vector $\mathbf{y}$ projected onto the dimensions specified by $v$. Notation $\mathbf{y}_{\bar{v}}$ is the complement, i.e., the vector of remaining candidate labels. The full label vector $\mathbf{y}$ can therefore be expressed as $\mathbf{y} = \{\mathbf{y}_v, \mathbf{y}_{\bar{v}}\}$, which means $\mathbf{y}$ is the concatenation of $\mathbf{y}_v$ and $\mathbf{y}_{\bar{v}}$. In PML, $\mathbf{y}_v$ is selected from $\mathbf{y}$.

Assume that $Q$ is a hypothetical predictive model with two parameters: $v$ representing which labels are selected and $w$ representing parameter used for prediction. *In order to identify the ground-truth labels from the candidate label set for PML, we can maximize the conditional likelihood of training dataset with respect to parameters {v, w}.* The conditional likelihood given all training examples $\mathcal{S} = \{(\mathbf{x}^i, \mathbf{y}^i) : i = 1, \ldots, n\}$ can be expressed as follows:

$$\mathcal{L}(v, w | \mathcal{S}) = \prod_{i=1}^{n} Q(\mathbf{y}_v^i | \mathbf{x}^i, w) \tag{1}$$

For convenience, we use the conditional log-likelihood corresponding to Eq. (1) as follows:

$$\ell(v, w | \mathcal{S}) = \frac{1}{n} \sum_{i=1}^{n} \log Q(\mathbf{y}_v^i | \mathbf{x}^i, w) \tag{2}$$

By multiplying and dividing classifier $Q$ by the distribution of identified ground-truth labels given features, i.e., $P(\mathbf{y}_v | \mathbf{x})$, we can re-write the above as follows:

$$\ell(v, w | \mathcal{S}) = \frac{1}{n} \sum_{i=1}^{n} \log \frac{Q(\mathbf{y}_v^i | \mathbf{x}^i, w)}{P(\mathbf{y}_v^i | \mathbf{x}^i)} + \frac{1}{n} \sum_{i=1}^{n} \log P(\mathbf{y}_v^i | \mathbf{x}^i) \tag{3}$$

Similarly, we can expand the second term in Eq. (3) by multiplying and dividing the probability $P(\mathbf{y} | \mathbf{x})$, and get the following formulation:

$$\ell(v, w | \mathcal{S}) = \frac{1}{n} \sum_{i=1}^{n} \log \frac{Q(\mathbf{y}_v^i | \mathbf{x}^i, w)}{P(\mathbf{y}_v^i | \mathbf{x}^i)} + \frac{1}{n} \sum_{i=1}^{n} \log \frac{P(\mathbf{y}_v^i | \mathbf{x}^i)}{P(\mathbf{y}^i | \mathbf{x}^i)} + \frac{1}{n} \sum_{i=1}^{n} \log P(\mathbf{y}^i | \mathbf{x}^i) \tag{4}$$

Furthermore, we use $\mathbb{E}_{(\mathbf{X}, \mathbf{Y})}$ to denote the expectation of the random variables $(\mathbf{X}, \mathbf{Y})$. When $n \to +\infty$, Eq. (4) can be reformated as follows:

$$\ell(v, w | \mathcal{S}) = \mathbb{E}_{(\mathbf{X}, \mathbf{Y})} \left\{ \log \frac{Q(\mathbf{y}_v | \mathbf{x}, w)}{P(\mathbf{y}_v | \mathbf{x})} \right\} + \mathbb{E}_{(\mathbf{X}, \mathbf{Y})} \left\{ \log \frac{P(\mathbf{y}_v | \mathbf{x})}{P(\mathbf{y} | \mathbf{x})} \right\} + \mathbb{E}_{(\mathbf{X}, \mathbf{Y})} \left\{ \log P(\mathbf{y} | \mathbf{x}) \right\} \tag{5}$$

The first term of Eq. (5) equals to the following KL-divergence:

$$\mathbb{E}_{(\mathbf{X}, \mathbf{Y})} \left\{ \log \frac{Q(\mathbf{y}_v | \mathbf{x}, w)}{P(\mathbf{y}_v | \mathbf{x})} \right\} = -D_{KL}(P(\mathbf{Y}_v | \mathbf{X}) || Q(\mathbf{Y}_v | \mathbf{X}, w)) \tag{6}$$

As $\mathbf{y} = \{\mathbf{y}_v, \mathbf{y}_{\bar{v}}\}$, the second term of Eq. (5) can be derived into the following formulation:

$$\begin{aligned}
\mathbb{E}_{(\mathbf{X}, \mathbf{Y})} \left\{ \log \frac{P(\mathbf{y}_v | \mathbf{x})}{P(\mathbf{y} | \mathbf{x})} \right\} &= -\mathbb{E}_{(\mathbf{X}, \mathbf{Y})} \left\{ \log \frac{P(\mathbf{y} | \mathbf{x})}{P(\mathbf{y}_v | \mathbf{x})} \right\} = -\sum_{(\mathbf{x}, \mathbf{y})} P(\mathbf{x}, \mathbf{y}) \log \frac{P(\mathbf{y}_{\bar{v}}, \mathbf{x} | \mathbf{y}_v)}{P(\mathbf{x} | \mathbf{y}_v)} \\
&= -\sum_{(\mathbf{x}, \mathbf{y})} P(\mathbf{x}, \mathbf{y}) \log \frac{P(\mathbf{y}_{\bar{v}}, \mathbf{x} | \mathbf{y}_v)}{P(\mathbf{x} | \mathbf{y}_v) P(\mathbf{y}_{\bar{v}} | \mathbf{y}_v)} - \sum_{(\mathbf{y}_{\bar{v}}, \mathbf{y}_v)} P(\mathbf{y}_{\bar{v}}, \mathbf{y}_v) \log P(\mathbf{y}_{\bar{v}} | \mathbf{y}_v) \\
&= -I(\mathbf{Y}_{\bar{v}}, \mathbf{X} | \mathbf{Y}_v) + H(\mathbf{Y}_{\bar{v}} | \mathbf{Y}_v) = -I(\mathbf{Y}_{\bar{v}}, \mathbf{X} | \mathbf{Y}_v)
\end{aligned} \tag{7}$$

In Eq. (7), $I(\mathbf{Y}_{\bar{v}}, \mathbf{X} | \mathbf{Y}_v)$ is the conditional mutual information and $H(\mathbf{Y}_{\bar{v}} | \mathbf{Y}_v)$ is the conditional entropy. The last equality holds as a result of the conditional entropy $H(\mathbf{Y}_{\bar{v}} | \mathbf{Y}_v) = 0$. This is because in PML setting, $\mathbf{Y}_v$ is selected from $\mathbf{Y}$, once $\mathbf{Y}_v$ is known, then $\mathbf{Y}$ is of course known in advance, meanwhile, $\mathbf{Y}_{\bar{v}}$ is complementary to $\mathbf{Y}_v$, thus, the uncertainty remaining in $\mathbf{Y}_{\bar{v}}$ is zero, i.e., $H(\mathbf{Y}_{\bar{v}} | \mathbf{Y}_v) = 0$. The detailed derivation of Eq. (7) can be found in the supplementary materials.

Moreover, the third term of Eq. (5) equals to the conditional entropy $-H(\mathbf{Y} | \mathbf{X})$, i.e.,

$$\frac{1}{n} \sum_{i=1}^{n} \log P(\mathbf{y}^i | \mathbf{x}^i) = -H(\mathbf{Y} | \mathbf{X}) \tag{8}$$

Combining together, we get the derived objective function as follows:

$$\ell(v, w | \mathcal{S}) = -D_{KL}(P(\mathbf{Y}_v | \mathbf{X}) || Q(\mathbf{Y}_v | \mathbf{X}, w)) - I(\mathbf{Y}_{\bar{v}}, \mathbf{X} | \mathbf{Y}_v) - H(\mathbf{Y} | \mathbf{X}) \tag{9}$$

The first term is a KL-divergence between the true and predicted ground-truth label distributions given features. The value of this term depends on how well the model $Q$ can approximate $P$. The second term is the conditional mutual information between the remaining candidate labels and the features, given the identified ground-truth labels. The value of this term depends solely on the choice of ground-truth labels, and will decrease as the selected ground-truth label set $\mathbf{Y}_v$ explains more about features $\mathbf{X}$, until eventually becoming zero when the remaining candidate labels $\mathbf{Y}_{\bar{v}}$ are all false positive labels, which contain no additional information about $X$. The last term is the conditional entropy of all candidate labels given features. This term quantifies the uncertainty remaining in the candidate labels after we know all features of training data, which is independent of parameters $v$ and $w$ as a constant.

# 4 Theoretical Analysis

To answer the first question:"Can the ground-truth labels be identified precisely?", we define an $\epsilon$-identifiable score function to decide whether a PML method can identify the ground-truth label precisely. Intuitively, the score of an identifiable score function increases when a ground-truth label is added to the identified label set compared to that of a false positive label. Thus, the $\epsilon$-identifiable score function can be defined as follows,

**Definition 1** ($\epsilon$-identifiable Score Function). *Let $\hat{S}$ be the identified label set; $y_k$ be a ground-truth label and $y_{\bar{k}}$ be a false positive label. Let $\epsilon$ be a non-negative constant. A score function $g : 2^{[q]} \to \mathbb{R}$ is said to be $\epsilon$-identifiable if the following property holds: for every $y_k \notin \hat{S}$ and $y_{\bar{k}} \notin \hat{S}$, we have $g(\hat{S} \cup \{y_k\}) - g(\hat{S} \cup \{y_{\bar{k}}\}) \geq \epsilon$.*

We will use this definition to derive the following theorem, which demonstrates that our objective function is an $\epsilon$-identifiable score function and is capable of identifying the ground-truth label from the candidate label set.

**Theorem 1.** *Let $y_k$ and $y_{\bar{k}}$ denote any unidentified ground-truth label and any false positive label respectively. Let $\mathbf{Y}_k$ and $\mathbf{Y}_{\bar{k}}$ represent the random variables of $y_k$ and $y_{\bar{k}}$ respectively, and that $y_k, y_{\bar{k}} \in \bar{v}$ and $y_k, y_{\bar{k}} \notin v$. Then, for any identified ground-truth label set $v$, our objective function $\ell(v, w|\mathcal{S})$ is an $\epsilon$-identifiable score function on condition that,*

$$I(\mathbf{Y}_k, \mathbf{X}) - I(\mathbf{Y}_k, \mathbf{Y}_v) + I(\mathbf{Y}_k, \mathbf{Y}_v|\mathbf{X}) - D_{KL1} \geq I(\mathbf{Y}_{\bar{k}}, \mathbf{X}) - I(\mathbf{Y}_{\bar{k}}, \mathbf{Y}_v) + I(\mathbf{Y}_{\bar{k}}, \mathbf{Y}_v|\mathbf{X}) - D_{KL2}$$

*where $D_{KL1} = D_{KL}\Big(P(\mathbf{Y}_v \cup \mathbf{Y}_k|\mathbf{X})||Q(\mathbf{Y}_v \cup \mathbf{Y}_k|\mathbf{X}, w)\Big)$*

*and $D_{KL2} = D_{KL}\Big(P(\mathbf{Y}_v \cup \mathbf{Y}_{\bar{k}}|\mathbf{X})||Q(\mathbf{Y}_v \cup \mathbf{Y}_{\bar{k}}|\mathbf{X}, w)\Big)$*

$$(10)$$

*Proof.*

$$g(\hat{S} \cup \{y_k\}) - g(\hat{S} \cup \{y_{\bar{k}}\}) = \Big\{ -I\Big(\mathbf{Y}_{\bar{v}}\backslash\mathbf{Y}_k, \mathbf{X}\big|\mathbf{Y}_v \cup \mathbf{Y}_k\Big) + I\Big(\mathbf{Y}_{\bar{v}}\backslash\mathbf{Y}_{\bar{k}}, \mathbf{X}\big|\mathbf{Y}_v \cup \mathbf{Y}_{\bar{k}}\Big)\Big\} +$$
$$\Big\{ -D_{KL}\Big(P(\mathbf{Y}_v \cup \mathbf{Y}_k|\mathbf{X})||Q(\mathbf{Y}_v \cup \mathbf{Y}_k|\mathbf{X}, w)\Big) + D_{KL}\Big(P(\mathbf{Y}_v \cup \mathbf{Y}_{\bar{k}}|\mathbf{X})||Q(\mathbf{Y}_v \cup \mathbf{Y}_{\bar{k}}|\mathbf{X}, w)\Big)\Big\}$$
$$(11)$$

Firstly, we derive the first part of Eq. (11), i.e., $-I\Big(\mathbf{Y}_{\bar{v}}\backslash\mathbf{Y}_k, \mathbf{X}\big|\mathbf{Y}_v \cup \mathbf{Y}_k\Big) + I\Big(\mathbf{Y}_{\bar{v}}\backslash\mathbf{Y}_{\bar{k}}, \mathbf{X}\big|\mathbf{Y}_v \cup \mathbf{Y}_{\bar{k}}\Big)$. Due to the chain rule that $I(\mathbf{Y}, \mathbf{X}) = I(\mathbf{Y}_v, \mathbf{X}) + I(\mathbf{Y}_{\bar{v}}, \mathbf{X}|\mathbf{Y}_v)$, we have

$$\begin{aligned} -I\Big(\mathbf{Y}_{\bar{v}}\backslash\mathbf{Y}_k, \mathbf{X}\big|\mathbf{Y}_v \cup \mathbf{Y}_k\Big) &= -\Big\{I(\mathbf{Y}, \mathbf{X}) - I(\mathbf{Y}_v \cup \mathbf{Y}_k, \mathbf{X})\Big\} = I(\mathbf{Y}_v \cup \mathbf{Y}_k, \mathbf{X}) - I(\mathbf{Y}, \mathbf{X}) \\ &= I(\mathbf{Y}_v, \mathbf{X}) + I(\mathbf{Y}_k, \mathbf{X}|\mathbf{Y}_v) - I(\mathbf{Y}, \mathbf{X}) \\ &= I(\mathbf{Y}_v, \mathbf{X}) + I(\mathbf{Y}_k, \mathbf{X}) - I(\mathbf{Y}_k, \mathbf{Y}_v) + I(\mathbf{Y}_k, \mathbf{Y}_v|\mathbf{X}) - I(\mathbf{Y}, \mathbf{X}) \end{aligned}$$
$$(12)$$

Similarly, we can get,

$$-I\Big(\mathbf{Y}_{\bar{v}}\backslash\mathbf{Y}_{\bar{k}}, \mathbf{X}\big|\mathbf{Y}_v \cup \mathbf{Y}_{\bar{k}}\Big) = I(\mathbf{Y}_v, \mathbf{X}) + I(\mathbf{Y}_{\bar{k}}, \mathbf{X}) - I(\mathbf{Y}_{\bar{k}}, \mathbf{Y}_v) + I(\mathbf{Y}_{\bar{k}}, \mathbf{Y}_v|\mathbf{X}) - I(\mathbf{Y}, \mathbf{X}) \quad (13)$$

Therefore, we have the following expression,

$$\begin{aligned} \text{The first part of } Eq. \text{ (11)} &= -I\Big(\mathbf{Y}_{\bar{v}}\backslash\mathbf{Y}_k, \mathbf{X}\big|\mathbf{Y}_v \cup \mathbf{Y}_k\Big) + I\Big(\mathbf{Y}_{\bar{v}}\backslash\mathbf{Y}_{\bar{k}}, \mathbf{X}\big|\mathbf{Y}_v \cup \mathbf{Y}_{\bar{k}}\Big) \\ &= \Big\{I(\mathbf{Y}_k, \mathbf{X}) - I(\mathbf{Y}_k, \mathbf{Y}_v) + I(\mathbf{Y}_k, \mathbf{Y}_v|\mathbf{X})\Big\} - \Big\{I(\mathbf{Y}_{\bar{k}}, \mathbf{X}) - I(\mathbf{Y}_{\bar{k}}, \mathbf{Y}_v) + I(\mathbf{Y}_{\bar{k}}, \mathbf{Y}_v|\mathbf{X})\Big\} \end{aligned}$$
$$(14)$$

where $I(\mathbf{Y}_k, \mathbf{X})$ is the mutual information between a ground-truth label $y_k$ and features, which means the correlation between the two items; $I(\mathbf{Y}_k, \mathbf{Y}_v)$ is the mutual information between a ground-truth label $y_k$ and the existing identified ground-truth label set, which means the redundancy between the two items; $I(\mathbf{Y}_k, \mathbf{Y}_v|\mathbf{X})$ is the feature-conditional redundancy, which means the good information still remaining between ground-truth label $y_k$ and the existing identified label set $v$ after features are known.

Thus, the interpretation of the first part of Eq. (14), i.e., $I(\mathbf{Y}_k, \mathbf{X}) - I(\mathbf{Y}_k, \mathbf{Y}_v) + I(\mathbf{Y}_k, \mathbf{Y}_v|\mathbf{X})$, is the overall correlation of a ground-truth label $y_k$ and features $\mathbf{x}$, which equals the correlation between a ground-truth label $y_k$ and features $\mathbf{x}$ minus the overall redundancy of label $y_k$ with existing identified label set, based on which we can evaluate the effectiveness to include the label $y_k$ into the ground-truth label set.

Similarly, the second part of Eq. (14) stands for the overall correlation between a false-positive label $y_{\bar{k}}$ and features $\mathbf{x}$, which equals the correlation between a false-positive label $y_{\bar{k}}$ and features $\mathbf{x}$ minus the overall redundancy of label $y_{\bar{k}}$ with existing identified label set.

It is naturally hold that the first part should be larger than the sencond part as the overall correlation between a ground-truth label $y_k$ and features ought to be stronger than that of a false-positive label $y_{\bar{k}}$ and features.

After that, we illustrate the second part of Eq. (11), i.e., $-D_{KL}\Big(P(\mathbf{Y}_v \cup \mathbf{Y}_k|\mathbf{X})||Q(\mathbf{Y}_v \cup \mathbf{Y}_k|\mathbf{X}, w)\Big) + D_{KL}\Big(P(\mathbf{Y}_v \cup \mathbf{Y}_{\bar{k}}|\mathbf{X})||Q(\mathbf{Y}_v \cup \mathbf{Y}_{\bar{k}}|\mathbf{X}, w)\Big)$. The interpretaion of $D_{KL}\Big(P(\mathbf{Y}_v \cup \mathbf{Y}_k|\mathbf{X})||Q(\mathbf{Y}_v \cup \mathbf{Y}_k|\mathbf{X}, w)\Big)$ is how well can the predictive model $Q$ approximate the true probability distribution $P$ when a ground-truth label $y_k$ is added to the existing identified label set $v$; that is, the information loss when using $Q$ to approximate $P$. The more precise the approximation is, the smaller is the KL-divergence value.

Similarly, $D_{KL}\Big(P(\mathbf{Y}_v \cup \mathbf{Y}_{\bar{k}}|\mathbf{X})||Q(\mathbf{Y}_v \cup \mathbf{Y}_{\bar{k}}|\mathbf{X}, w)\Big)$ can be interpreted as how well can the predictive model $Q$ approximate the true probability distribution $P$ when a a false-positive label $y_{\bar{k}}$ is added to the existing identified label set $v$. The more imprecise the approximation is, the bigger is the KL-divergence value.

It is naturally hold that the KL-divergence value should be smaller when the ground-truth label $y_k$ is added to the existing identified label set $v$ than that of the false-positive label $y_{\bar{k}}$, that is, $D_{KL}\Big(P(\mathbf{Y}_v \cup \mathbf{Y}_{\bar{k}}|\mathbf{X})||Q(\mathbf{Y}_v \cup \mathbf{Y}_{\bar{k}}|\mathbf{X}, w)\Big) \geq D_{KL}\Big(P(\mathbf{Y}_v \cup \mathbf{Y}_k|\mathbf{X})||Q(\mathbf{Y}_v \cup \mathbf{Y}_k|\mathbf{X}, w)\Big)$.

Therefore, combined with the derivation of the first part of Eq. (11), we can get the condition for our objective function $\ell(v, w|\mathcal{S})$ to be an $\epsilon$-identifiable score function as follows,

$$I(\mathbf{Y}_k, \mathbf{X}) - I(\mathbf{Y}_k, \mathbf{Y}_v) + I(\mathbf{Y}_k, \mathbf{Y}_v|\mathbf{X}) - D_{KL1} \geq I(\mathbf{Y}_{\bar{k}}, \mathbf{X}) - I(\mathbf{Y}_{\bar{k}}, \mathbf{Y}_v) + I(\mathbf{Y}_{\bar{k}}, \mathbf{Y}_v|\mathbf{X}) - D_{KL2}$$

where $D_{KL1} = D_{KL}\Big(P(\mathbf{Y}_v \cup \mathbf{Y}_k|\mathbf{X})||Q(\mathbf{Y}_v \cup \mathbf{Y}_k|\mathbf{X}, w)\Big)$ and $D_{KL2} = D_{KL}\Big(P(\mathbf{Y}_v \cup \mathbf{Y}_{\bar{k}}|\mathbf{X})||Q(\mathbf{Y}_v \cup \mathbf{Y}_{\bar{k}}|\mathbf{X}, w)\Big)$, which concludes the proof. $\square$

**Remark.** The condition of Eq. (10) requires that the overall correlation between a ground-truth label $y_k$ and features should be stronger than the overall correlation between a false-positive label $y_{\bar{k}}$ and features. From the above theoretical analysis, we can see that the ground-truth labels can be identified correctly by the proposed MILI-PML under given conditions and assumptions in Theorem 1.

## 5 Optimization

To answer the second question:"How can the ground-truth labels be found?", we design an efficient algorithm to optimize the proposed MILI-PML.

The goal of partial multi-label learning (PML) is to maximize $\ell(v, w|\mathcal{S})$ of Eq. (9), which is equivalent to minimizing $-\ell(v, w|\mathcal{S})$. Consequently, we can rewrite the objective function of MILI-PML as follows:

$$-\ell(v, w|\mathcal{S}) = D_{KL}(P(\mathbf{Y}_v|\mathbf{X})||Q(\mathbf{Y}_v|\mathbf{X}, w)) + I(\mathbf{Y}_{\bar{v}}, \mathbf{X}|\mathbf{Y}_v) + H(\mathbf{Y}|\mathbf{X}) \tag{15}$$

As the last term $H(\mathbf{Y}|\mathbf{X})$ is a constant, we only need to consider optimizing the first term and the second term together in Eq. (15). Thus, minimizing $-\ell(v, w|\mathcal{S})$ is equivalent to minimizing $D_{KL}(P(\mathbf{Y}_v|\mathbf{X})||Q(\mathbf{Y}_v|\mathbf{X}, w)) + I(\mathbf{Y}_{\bar{v}}, \mathbf{X}|\mathbf{Y}_v)$, that is,

$$\arg\min_{v, w} -\ell(v, w|\mathcal{S}) = \arg\min_{v, w} D_{KL}(P(\mathbf{Y}_v|\mathbf{X})||Q(\mathbf{Y}_v|\mathbf{X}, w)) + I(\mathbf{Y}_{\bar{v}}, \mathbf{X}|\mathbf{Y}_v) \tag{16}$$

---

**Algorithm 1** Alternating Optimization (MILI Algorithm)

---

**Goal:** Solve the Optimization Problem of Eq. (16) : identifying the ground-truth label set $v$ as well as training a predictive model parameter $w$;

**Input:** training data $\mathcal{S} = \{(\mathbf{x}^i, \mathbf{y}^i) : i = 1, \dots, n\}$;

**Output:** the predictive model parameter $w$.

1: Initialize the ground-truth label set $v^i$ for each instance by randomly choosing from the corresponding candidate label set $\bar{v}^i$; $t = 0$; $F^0(v, w) = 0$;
2: **repeat**
3:     Fix $v$, update $w$ in Eq. (17) according to Gradient Descent Algorithm.
4:     Fix $w$, update $v$ in Eq. (19) according to Algorithm 2;
5:     t = t+1;
6:     Calculate $F^{t+1}(v, w)$;
7: **until** $\frac{|F^{t+1}(v,w) - F^t(v,w)|}{|F^t(v,w)|} < \delta$;
8: Output the optimal solution $v^i$ for each instance $\mathbf{x}^i$, $i \in \{1, \dots, n\}$ and the predictive model parameter $w$.

---

We design an alternating optimization algorithm called Mutual Information Label Identification Algorithm (i.e., MILI Algorithm) to optimize the two parameters $v$ (the ground-truth label set parameter) and $w$ (the predictive model parameter) in Eq. (16) by fixing one and updating the other. Complete procedures of MILI Algorithm are summarized in Algorithm 1.

For convenience, we denote the objective function in Eq. (16) as $F(v, w) = D_{KL}(P(\mathbf{Y}_v|\mathbf{X})||Q(\mathbf{Y}_v|\mathbf{X}, w)) + I(\mathbf{Y}_{\bar{v}}, \mathbf{X}|\mathbf{Y}_v)$, which is used to set the stopping creteria in Algorithm 1.

Specifically, when $v$ is fixed to update $w$, Eq. (16) is equivalent to optimize the following problem as the second term $I(\mathbf{Y}_{\bar{v}}, \mathbf{X}|\mathbf{Y}_v)$ becomes a constant.

$$\arg \min_w \mathbb{E}_{(\mathbf{X},\mathbf{Y})} \log P(\mathbf{y}_v|\mathbf{x}) - \mathbb{E}_{(\mathbf{X},\mathbf{Y})} \log Q(\mathbf{y}_v|\mathbf{x}, w) = \arg \min_w - \mathbb{E}_{(\mathbf{X},\mathbf{Y})} \log Q(\mathbf{y}_v|\mathbf{x}, w) \quad (17)$$

Moreover, we can use the sample estimation to approximate $\mathbb{E}_{(\mathbf{X},\mathbf{Y})} \log Q(\mathbf{y}_v|\mathbf{x}, w)$ based on the Law of Large Numbers as follows:

$$\mathbb{E}_{(\mathbf{X},\mathbf{Y})} \log Q(\mathbf{y}_v|\mathbf{x}, w) \approx \frac{1}{n} \sum_{i=1}^{n} \log Q(\mathbf{y}_v^i|\mathbf{x}^i, w) \quad (18)$$

To this end, we can employ the gradient descent algorithm to optimize Eq. (17).

Next, we fix $w$ to update $v$. The first term of Eq. (16) can be decomposed to be $\mathbb{E}_{(\mathbf{X},\mathbf{Y})} \log P(\mathbf{y}_v|\mathbf{x}) - \mathbb{E}_{(\mathbf{X},\mathbf{Y})} \log Q(\mathbf{y}_v|\mathbf{x})$. Meanwhile, due to the chain rule that $I(\mathbf{Y}, \mathbf{X}) = I(\mathbf{Y}_v, \mathbf{X}) + I(\mathbf{Y}_{\bar{v}}, \mathbf{X}|\mathbf{Y}_v)$, minimizing $I(\mathbf{Y}_{\bar{v}}, \mathbf{X}|\mathbf{Y}_v)$ is equivalent to maximizing $I(\mathbf{Y}_v, \mathbf{X})$. Therefore, Eq. (16) can be reformulated as follows:

$$\arg \max_v - \mathbb{E}_{(\mathbf{X},\mathbf{Y})} \log P(\mathbf{y}_v|\mathbf{x}) + \mathbb{E}_{(\mathbf{X},\mathbf{Y})} \log Q(\mathbf{y}_v|\mathbf{x}) + I(\mathbf{Y}_v, \mathbf{X}) \quad (19)$$

Moreover, we can identify ground-truth labels $y_k$ from $\bar{v}$ one by one and add to $v$. Thus, Eq. (19) is equivalent to the following formulation:

$$\arg \max_{y_k \in \bar{v}} - \mathbb{E}_{(\mathbf{X},\mathbf{Y})} \log P(y_k|\mathbf{x}, \mathbf{y}_v) + \mathbb{E}_{(\mathbf{X},\mathbf{Y})} \log Q(y_k|\mathbf{x}, \mathbf{y}_v) + I(\mathbf{Y}_k, \mathbf{X}|\mathbf{Y}_v) \quad (20)$$

In practice, we can use the empirical distribution and sample estimation to approximate the three terms in Eq. (20) based on the Law of Large Numbers. To this end, we can employ greedy strategy to optimize Eq. (19). Procedures of this implementation can be found in Algorithm 2.

Here, we explain some main steps of Algorithm 1. Step 2–7 summarizes the core iteration procedures, among which Step 3 and Step 4 are the alternating strategy to optimize the two parameters $w$ and $v$, respectively. At Step 6, the objective function value $F(v, w)$ is calculated. Finally, the stopping condition is set to $\delta$-optimal and we choose $\delta$ to be $10^{-5}$ in practice (Step 7).

---

**Algorithm 2** Fix $w$, update $v$.

---

**Goal:** Solve the Optimization Problem of Eq. (19).
**Input:** training data $\mathcal{S} = \{(\mathbf{x}^i, \mathbf{y}^i) : i = 1, \ldots, n\}$;
**Output:** the ground-truth label set $v^i$ for each instance $i \in \{1, \ldots, n\}$.

1: Initialize $t = 0$; $v_0^i = \emptyset$ and let $\bar{v}_0^i$ be the initial candidate label set for each instance $\mathbf{x}^i$, $i \in \{1, \ldots, n\}$;
2: **while** $I(\mathbf{Y}_k, \mathbf{X}|\mathbf{Y}_{v_t^i}) > \varepsilon$ **do**
3:     Identifying the ground-truth label $y_k^i$ by Eq. (20);
4:     Update the ground-truth label set by $v_{t+1}^i = v_t^i \cup y_k^i$;
5:     Update the remaining candidate label set by $\bar{v}_{t+1}^i = \bar{v}_t^i \backslash y_k^i$;
6:     $t = t + 1$;
7: **end while**
8: Output $v^i$ by $v^i = v_{\hat{t}}^i$.

---

## 6 Experiments

In this section, we conduct experiments to evaluate the classification performance of the proposed MILI-PML and compare it with six state-of-the-art PML methods.

### 6.1 Datasets

Experiments are conducted on six synthetic PML datasets[1] and four real-world PML datasets (i.e. YeastBP [16], Music-emotion [19], Music-style [19], MIRFlickr [19, 14] of different scales, the characteristics of which are summarized in the supplementary materials. For real-world PML datasets, candidate labels are collected from web users which are further examined by human labelers to specify the ground-truth labels. For synthetic datasets, given the configuration strategy over multi-label datasets in [1, 14], we construct the candidate label set by randomly choosing irrelevant labels together with the ground-truth labels (GLs) for each multi-label instance. Specifically, different settings are considered by varying the average number of candidate labels (#CLs (avg.)) for each multi-label dataset. Accordingly, in this paper, we generate thirty synthetic PML datasets. For brevity, we report the detailed results of two configurations for each dataset, i.e. avg. #CLs (avg.) being 7 and 11 for Enron, Corel5k, Eurlex-sm; 9 and 13 for Eurlex-ed and Mediamill; 45 and 65 for CAL500.

Table 1: Experimental results of the proposed MILI-PML with six state-of-the-art PML baselines on four real-world as well as six synthetic PML datasets in terms of **ranking loss**. The best result (the smaller the better) is *in bold*.

| Dataset | #CLs (avg.) | MILI-PML | PAR-VLS | DRAMA | PML-lc | fPML | PML-LRS | MUSER |
|---|---|---|---|---|---|---|---|---|
| YeastBP | 30.43 | .357±.017 | .436±.032 | .407±.021 | .382±.036 | .414±.012 | .418±.031 | **.341±.015** |
| Music-emotion | 5.29 | **.171±.008** | .260±.005 | .218±.013 | .254±.006 | .352±.015 | .281±.005 | .189±.021 |
| Music-style | 6.04 | **.162±.007** | .182±.021 | .178±.013 | .267±.013 | .238±.019 | .179±.007 | .171±.006 |
| MIRFlickr | 3.35 | **.081±.005** | .203±.007 | .189±.010 | .146±.008 | .163±.025 | .107±.002 | .093±.006 |
| Enron | 7 | **.082±.002** | .297±.007 | .194±.012 | .338±.004 | .128±.017 | .207±.021 | .114±.003 |
| | 11 | **.096±.004** | .312±.006 | .210±.015 | .341±.002 | .127±.008 | .215±.017 | .123±.004 |
| Corel5k | 7 | **.012±.015** | .345±.070 | .193±.052 | .161±.013 | .132±.027 | .193±.016 | .015±.004 |
| | 11 | **.015±.007** | .383±.056 | .201±.065 | .171±.014 | .138±.013 | .202±.021 | .017±.005 |
| Eurlex-sm | 7 | **.041±.003** | .058±.014 | .051±.007 | .079±.017 | .313±.072 | .183±.006 | .043±.005 |
| | 11 | **.047±.006** | .067±.009 | .063±.016 | .081±.016 | .368±.016 | .197±.008 | .049±.003 |
| Eurlex-ed | 9 | .045±.006 | .064±.013 | .068±.008 | .083±.016 | .328±.009 | .198±.008 | **.041±.003** |
| | 13 | .062±.017 | .069±.025 | .071±.013 | .092±.018 | .412±.024 | .217±.010 | **.043±.002** |
| CAL500 | 45 | **.175±.007** | .353±.014 | .235±.007 | .316±.008 | .268±.015 | .281±.013 | .179±.024 |
| | 65 | **.203±.027** | .471±.012 | .317±.016 | .365±.014 | .287±.019 | .347±.016 | .213±.021 |
| Mediamill | 9 | **.127±.009** | .135±.008 | .201±.007 | .216±.015 | .203±.007 | .193±.107 | .173±.002 |
| | 13 | **.151±.004** | .198±.004 | .301±.012 | .316±.009 | .225±.008 | .212±.023 | .187±.021 |

### 6.2 Baselines

We compare the proposed MILI-PML method with the six state-of-the-art PML approaches.

---

[1]http://mulan.sourceforge.net/datasets-mlc.html

Table 2: Experimental results of the proposed MILI-PML with six state-of-the-art PML baselines on four real-world as well as six synthetic PML datasets in terms of **average precision**. The best result (the larger the better) is *in bold*.

| Dataset | #CLs (avg.) | MILI-PML | PAR-VLS | DRAMA | PML-lc | fPML | PML-LRS | MUSER |
|---|---|---|---|---|---|---|---|---|
| YeastBP | 30.43 | .151±.027 | .082±.031 | .083±.017 | .140±.035 | .096±.021 | .085±.023 | **.154±.031** |
| Music-emotion | 5.29 | **.603±.018** | .527±.006 | .582±.012 | .541±.023 | .538±.015 | .516±.014 | .598±.033 |
| Music-style | 6.04 | **.722±.017** | .717±.031 | .693±.013 | .627±.010 | .659±.017 | .716±.018 | .718±.013 |
| MIRFlickr | 3.35 | **.807±.015** | .685±.017 | .707±.014 | .743±.018 | .731±.015 | .796±.012 | .801±.016 |
| Enron | 7 | **.784±.002** | .601±.006 | .613±.002 | .679±.003 | .751±.012 | .782±.011 | .771±.003 |
| | 11 | **.683±.005** | .587±.006 | .556±.012 | .660±.004 | .670±.006 | .683±.007 | .681±.005 |
| Corel5k | 7 | **.283±.012** | .205±.012 | .235±.014 | .253±.023 | .264±.017 | .237±.013 | .280±.003 |
| | 11 | **.279±.007** | .196±.036 | .218±.025 | .226±.013 | .258±.015 | .217±.011 | .276±.015 |
| Eurlex-sm | 7 | **.752±.013** | .741±.024 | .744±.017 | .718±.037 | .685±.022 | .699±.016 | .751±.025 |
| | 11 | **.749±.016** | .721±.016 | .728±.013 | .716±.006 | .628±.027 | .615±.017 | .748±.023 |
| Eurlex-ed | 9 | .753±.016 | .735±.013 | .727±.016 | .719±.024 | .686±.010 | .696±.015 | **.755±.023** |
| | 13 | .748±.027 | .728±.015 | .725±.023 | .715±.017 | .668±.014 | .681±.016 | **.752±.012** |
| CAL500 | 45 | **.626±.017** | .446±.024 | .563±.027 | .581±.018 | .531±.025 | .516±.023 | .620±.014 |
| | 65 | **.585±.021** | .432±.012 | .481±.046 | .434±.015 | .412±.022 | .448±.014 | .479±.018 |
| Mediamill | 9 | **.769±.019** | .756±.018 | .698±.017 | .685±.025 | .695±.017 | .689±.010 | .716±.012 |
| | 13 | **.728±.014** | .699±.024 | .687±.014 | .685±.019 | .674±.018 | .686±.013 | .702±.021 |

Table 3: Experimental results of the proposed MILI-PML with six state-of-the-art PML baselines on four real-world as well as six synthetic PML datasets in terms of **coverage**. The best result (the smaller the better) is *in bold*.

| Dataset | #CLs (avg.) | MILI-PML | PAR-VLS | DRAMA | PML-lc | fPML | PML-LRS | MUSER |
|---|---|---|---|---|---|---|---|---|
| YeastBP | 30.43 | .422±.024 | .731±.032 | **.417±.025** | .489±.036 | .494±.012 | .668±.031 | .431±.015 |
| Music-emotion | 5.29 | **.373±.008** | .461±.005 | .428±.013 | .434±.006 | .451±.015 | .483±.005 | .387±.021 |
| Music-style | 6.04 | **.182±.007** | .208±.021 | .217±.032 | .367±.013 | .238±.019 | .279±.007 | .216±.016 |
| MIRFlickr | 3.35 | **.208±.005** | .263±.007 | .289±.010 | .248±.008 | .267±.025 | .287±.002 | .233±.016 |
| Enron | 7 | **.281±.022** | .397±.007 | .394±.012 | .431±.004 | .421±.017 | .415±.021 | .314±.033 |
| | 11 | **.293±.014** | .416±.026 | .415±.015 | .446±.022 | .426±.018 | .417±.017 | .323±.024 |
| Corel5k | 7 | **.225±.017** | .415±.070 | .439±.052 | .361±.011 | .342±.017 | .293±.015 | .238±.014 |
| | 11 | **.275±.027** | .463±.026 | .451±.025 | .373±.014 | .338±.023 | .372±.021 | .278±.005 |
| Eurlex-sm | 7 | **.152±.003** | .162±.014 | .171±.007 | .358±.017 | .363±.012 | .183±.006 | .174±.005 |
| | 11 | **.187±.006** | .263±.009 | .257±.016 | .471±.016 | .426±.016 | .267±.008 | .219±.003 |
| Eurlex-ed | 9 | .239±.016 | .244±.013 | .278±.018 | .483±.026 | .386±.029 | **.238±.018** | .241±.013 |
| | 13 | .246±.017 | .279±.025 | .277±.013 | .492±.018 | .412±.024 | .247±.010 | **.245±.012** |
| CAL500 | 45 | **.585±.027** | .872±.024 | .838±.027 | .936±.038 | .875±.025 | .865±.033 | .679±.027 |
| | 65 | **.593±.027** | .953±.012 | .917±.016 | .955±.014 | .881±.019 | .878±.016 | .712±.012 |
| Mediamill | 9 | **.205±.019** | .211±.018 | .298±.024 | .317±.012 | .217±.013 | .291±.019 | .212±.022 |
| | 13 | **.215±.021** | .321±.027 | .362±.015 | .376±.007 | .266±.017 | .322±.021 | .257±.010 |

- *PARTICLE* [14]: An identifying method, which tries to extract credible labels with high-confidence values by label propagation procedure, and then trains classifiers by applying two exisiting multi-label models, which are PAR-VLS and PAR-MAP for short. Here, we choose PAR-VLS for comparison.

- *DRAMA* [15]: An identifying method, which tries to get the reliable labels with high-confidence by considering the structure of feature space, and then induces a gradient boosting model to train classifiers.

- *PML-fp and PML-lc* [1]: An embedding method, which attempts to figure out the label confidence by minimizing the ranking loss and exploiting data structure information with two models: one considering feature prototype (i.e., PML-fp) and the other considering label correlations (i.e., PML-lc). Here, we choose PML-lc for comparison.

- *fPML* [16]: An embedding method, which figures out the label confidence by adopting a feature and label coherent matrix to factorize the original matrix for prediction.

- *PML-LRS* [17]: An embedding method, which utilizes low-rank and sparse decomposition to capture the ground-truth label matrix and irrelevant label matrix from the observed candidate label matrix.

- *MUSER* [18]: An embedding method, which considers redundant labels together with noisy features and figures out the label confidence via optimizing correlation matrix.

For all PML baselines, we set the trade-off parameters as suggested in the original papers. Details can be found in the supplementary materials.

For MILI-PML, we employ the simple Binary Relevance [20, 21] as the predictive classifier $Q$ in the MILI-PML model. In addition, LIBLINEAR [22] with L2-regularized square hinge loss is also employed to train the binary classifiers in Binary Relevance (BR).

*Evaluation metrics*: We employ five widely-used multi-label metrics including ranking loss, hamming loss, one error, coverage, and average precision to evaluate the performance of all methods. More details about these evaluation metrics can be found in [23, 24, 25]. Besides, on each dataset, five-fold cross validation is performed where the mean metric value as well as standard deviation are recorded for each comparing method.

### 6.3 Experimental Results

Due to the page limit, we only report the performance comparisons of the proposed MILI-PML with six state-of-the-art PML methods on four real-world datasets and six synthetic datasets in terms of ranking loss, average precision and coverage in Tables 1, 2, 3 respectively. Besides, similar results can be observed in other metrics and we report the detailed results of hamming loss and one error metrics in the supplementary materials. From the overall results, we make the following observations:

- The proposed MILI-PML consistently outperforms all baselines on most real-world datasets, like Music-emotion, Music-style and MIRFlickr datasets, while is comparable to the best performance on YeastBP dataset. For example, MILI-PML is comparable to MUSER in terms of ranking loss and average precision, while comparable to DRAMA in terms of coverage.

- MILI-PML is superior to all baselines on most synthetic datasets, like Enron, Corel5k, Eurlex-sm, CAL500 and Mediamill while is comparable to the best performance on Eurlex-ed dataset. Specifically, MILI-PML is comparable to MUSER in terms of ranking loss and average precision, comparable to PML-LRS and MUSER in terms of coverage.

- It is obvious to observe that MILI-PML performs the best on most real-world as well as most synthetic datasets, except on YeastBP and Eurlex-ed. This is maybe because the number of feature and class combinations is relatively large compared with the number of instances in YeastBP and Eurlex-ed datasets, which increases the difficulty for identifying the ground-truth labels.

- The overall results demonstrate the superiorities of MILI-PML, which mainly owes to the natural consideration of the relevancy between features and labels as well as the overall redundancy of including an identified label into the ground-truth label set in the derived MILI-PML. This aligns with our theoretical analysis that MILI-PML is $\epsilon$-identifiable and capable of identifying the ground-truth labels correctly on the given conditions and assumptions of the data distribution.

## 7 Conclusion

This paper provides a new insight into partial multi-label learning problem from the perspective of mutual information. Instead of using hand-made heuristic modeling, we propose a novel method called MILI-PML, which is derived from a clear probabilistic formulation directly and can be easily interpreted theoretically by information theory as a result of naturally incorporating the relevancy considerations. Moreover, we present two fundamental issues about the assumption of existing PML research. To answer the first question "CAN", we define an $\epsilon$-identifiable score function to evaluate whether a PML method is workable for identifying the ground-truth labels precisely. Furthermore, we demonstrate that the derived MILI-PML is $\epsilon$-identifiable under certain assumptions and conditions theoretically. To answer the second question "HOW", we design an efficient optimization algorithm called MILI Algorithm to optimize the objective function. Extensive experiments are conducted on six synthetic PML datasets and four real-world PML datasets. The experimental results clearly demonstrate that the proposed MILI-PML consistently outperforms the state-of-the-art methods in terms of five widely-used multi-label metrics including ranking loss, hamming loss, one error, coverage, and average precision.

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
