# Understanding Partial Multi-label Learning via Mutual Information (Supplementary)

**Xiuwen Gong** [†,‡], **Dong Yuan**[†], **Wei Bao** [†]

[†] Faculty of Engineering, The University of Sydney
[‡] Hunan Huishiwei Intelligent Technology Co., Ltd.
{xiuwen.gong, dong.yuan, wei.bao}@sydney.edu.au

## Abstract

In this supplementary file, we present the derivation of Eq. (7), the datasets, the remaining configurations and experimental results.

## 1 Derivation of Eq. (7) in main paper

$$
\begin{aligned}
\mathbb{E}_{(\mathbf{X},\mathbf{Y})}\Big\{\log\frac{P(\mathbf{y}_v|\mathbf{x})}{P(\mathbf{y}|\mathbf{x})}\Big\} &= -\mathbb{E}_{(\mathbf{X},\mathbf{Y})}\Big\{\log\frac{P(\mathbf{y}|\mathbf{x})}{P(\mathbf{y}_v|\mathbf{x})}\Big\} \\
&= -\sum_{(\mathbf{x},\mathbf{y})} P(\mathbf{x},\mathbf{y})\log\frac{P(\mathbf{y}_v,\mathbf{y}_{\bar{v}}|\mathbf{x})}{P(\mathbf{y}_v|\mathbf{x})} \\
&= -\sum_{(\mathbf{x},\mathbf{y})} P(\mathbf{x},\mathbf{y})\log\frac{P(\mathbf{y}_{\bar{v}},\mathbf{y}_v,\mathbf{x})}{P(\mathbf{y}_v,\mathbf{x})} \\
&= -\sum_{(\mathbf{x},\mathbf{y})} P(\mathbf{x},\mathbf{y})\log\frac{P(\mathbf{y}_{\bar{v}},\mathbf{x}|\mathbf{y}_v)}{P(\mathbf{x}|\mathbf{y}_v)} \\
&= -\sum_{(\mathbf{x},\mathbf{y})} P(\mathbf{x},\mathbf{y})\log\frac{P(\mathbf{y}_{\bar{v}},\mathbf{x}|\mathbf{y}_v)}{P(\mathbf{x}|\mathbf{y}_v)}\frac{P(\mathbf{y}_{\bar{v}}|\mathbf{y}_v)}{P(\mathbf{y}_{\bar{v}}|\mathbf{y}_v)} \\
&= -\sum_{(\mathbf{x},\mathbf{y})} P(\mathbf{x},\mathbf{y})\log\frac{P(\mathbf{y}_{\bar{v}},\mathbf{x}|\mathbf{y}_v)}{P(\mathbf{x}|\mathbf{y}_v)P(\mathbf{y}_{\bar{v}}|\mathbf{y}_v)} \\
&\quad - \sum_{(\mathbf{x},\mathbf{y})} P(\mathbf{x},\mathbf{y})\log P(\mathbf{y}_{\bar{v}}|\mathbf{y}_v) \\
&= -\sum_{(\mathbf{x},\mathbf{y})} P(\mathbf{x},\mathbf{y})\log\frac{P(\mathbf{y}_{\bar{v}},\mathbf{x}|\mathbf{y}_v)}{P(\mathbf{x}|\mathbf{y}_v)P(\mathbf{y}_{\bar{v}}|\mathbf{y}_v)} \\
&\quad - \sum_{(\mathbf{y}_{\bar{v}},\mathbf{y}_v)} P(\mathbf{y}_{\bar{v}},\mathbf{y}_v)\log P(\mathbf{y}_{\bar{v}}|\mathbf{y}_v) \\
&= -I(\mathbf{Y}_{\bar{v}},\mathbf{X}|\mathbf{Y}_v) + H(\mathbf{Y}_{\bar{v}}|\mathbf{Y}_v) \\
&= -I(\mathbf{Y}_{\bar{v}},\mathbf{X}|\mathbf{Y}_v)
\end{aligned}
\tag{1}
$$

35th Conference on Neural Information Processing Systems (NeurIPS 2021).

Table 1: Statistics of real-world PML datasets.

| Datasets | #Instances | #Features | #Classes | #CLs (avg.) |
|---|---|---|---|---|
| YeastBP | 560 | 5548 | 217 | 30.43 |
| Music-emotion | 6833 | 98 | 11 | 5.29 |
| Music-style | 6839 | 98 | 10 | 6.04 |
| MIRFlickr | 10433 | 100 | 7 | 3.35 |

Table 2: Statistics of synthetic PML datasets.

| Datasets | #Instances | #Features | #Classes | #GLs (avg.) | #CLs (avg.) | Domain |
|---|---|---|---|---|---|---|
| Enron | 1702 | 1001 | 53 | 3.38 | 5, 7, 9, 11, 13 | text |
| Corel5k | 5000 | 499 | 374 | 3.52 | 5, 7, 9, 11, 13 | image |
| Eurlex-sm | 19348 | 5000 | 201 | 2.21 | 5, 7, 9, 11, 13 | text |
| Eurlex-ed | 19348 | 5000 | 3993 | 5.31 | 7, 9, 11, 13, 15 | text |
| CAL500 | 502 | 68 | 174 | 26.04 | 35, 45, 55, 65, 75 | music |
| Mediamill | 43907 | 120 | 101 | 4.38 | 7, 9, 11, 13, 15 | video |

## 2 Datasets

The four real-world PML datasets and six synthetic PML datasets are summarized in Table 1 and Table 2 respectively.

## 3 Configurations

For all PML baselines, we set the trade-off parameters as suggested in the original papers. i.e., PAR-VLS and PAR-MAP: trade-off parameter $\alpha = 0.95$, credible label elicitation threshold $thr = 0.9$ and the number of neighours $k = 10$; DRAMA: $\delta_1 = 0.01$ and $\delta_2 = 1/0.5$; PML-fp and PML-lc: $C_1 = 1$, $C_2$ is chose from $\{1, 2, \ldots, 10\}$ and $C_3$ is chose from $\{1, 10, \ldots, 100\}$ with five-fold cross validation; fPML: $\lambda_2 = 1$; PML-LRS: trade-off parameters are set as $\gamma = 0.01$, $\beta = 0.1$ and $\eta = 1$; MUSER: $\alpha, \beta, \gamma$ are chosen from $\{10^{-3}, \ldots, 10^3\}$ with a grid search manner. Libsvm is used as the binary learning algorithm for PARTICLE.

## 4 Experimental Results