# OpenReview forum: "Understanding Partial Multi-Label Learning via Mutual Information"
_NeurIPS.cc/2021/Conference — NeurIPS 2021 Poster_

### Official Review · Reviewer_9mLo · 2021-07-12

**Rating:** 8
**Confidence:** 4

**Summary:**

Partial multi-label learning has appeared to be more useful in many applications. This paper first investigates some basic problems existed in the current research, and proposes a new method based on the mutual information, which appears to be interesting to the community.

**Limitations And Societal Impact:**

- Some state-of-the-art partial multi-label references are missing, such as 1) Partial Multi-Label Learning with Label Distribution 2) Noisy label tolerance: A new perspective of Partial Multi-Label Learning 3) Partial multi-label learning with mutual teaching.

- The explanation of Theorem 1 is weak; the author should provide more explanations.

- Can the author do the experiments on the image data set?


**Main Review:**

- This paper first investigates two basic issues existed in the current research of partial multi-label learning and provides the solid technical solutions with theoretical guarantee.

- A new method based on mutual information is proposed in this paper. The motivation and derivation are clear. The idea is interesting, which could motivate the community to bring some new tools from other domains to address partial multi-label learning tasks.

- The extensive experiments are conducted on various data sets to verify the superior performance of the proposed method.


**Time Spent Reviewing:**

1 hours

---

> ### Author Response · Authors · 2021-08-09
> **Response**
>
> Thanks for your constructive comments and suggestions. We address your main concerns as follows:
>
> Question: Some state-of-the-art partial multi-label references are missing…
>
> Answer: We would like to add the above missing references in the revised manuscript.
>
>
> Question: The explanation of Theorem 1 is weak; the author should provide more explanations.
>
> Answer: Theorem 1 introduces the condition for the proposed MILI-PML that the overall correlation between a ground-truth label and features should be stronger than the overall correlation between a false-positive label and features. When the condition holds, the objective function of the proposed MILI-PML model in Eq.(9) is an identifiable score function, which means MILI-PML is workable for separating the ground-truth labels from candidate label set.
>
>
> Question: Can the author do the experiments on the image data set?
>
> Answer: Yes, we have done experiments on image data set, such as Corel5k, MIRFlickr. From the results in Tables 3–5, we can see that the proposed MILI-PML performs the best in terms of ranking loss, average precision, and coverage on the image datasets.

---

### Official Review · Reviewer_JowG · 2021-07-12

**Rating:** 6
**Confidence:** 3

**Summary:**

The paper addresses the problem setting of partial multi-label learning (PML). The task is to select the ground-truth labels from a set of label candidates and disregard false-positive or noisy labels that may occur in the labelling process. The authors propose MILI-PML that exploits dependencies between labels and features for selection and relies on the assumption that ground-truth labels have a higher overall dependency between labels and features than false-positive ones. The paper provides theoretical analysis to show that the model can identify ground-truth labels from a candidate set correctly and propose a corresponding optimization procedure. Further, the authors conduct experiments to demonstrate the effectiveness of MILI-PML and claim that the method achieves superior performance compared to other SOTA methods.

** Thanks for the response to this review.

**Limitations And Societal Impact:**

The paper partly addresses the limitations of the proposed method by stating assumptions that need to be fulfilled. It does not address the potential negative social impacts of the application of the proposed method. The authors state in the theoretical analysis that the algorithm only works under given conditions and certain assumptions. Limitations due to configurations (for example, which predictive model $Q$ is used) are not described. The experimental part does not examine whether the central assumptions about dependencies in the data hold and to what extent MILI-PML exploits them. I would appreciate it if the authors would comment in the paper on the potential impacts of their method. For example, I would think the method could be of interest in the context of automatic filtering of "noisy" or "false" labels and could lead to bias or discrimination depending on the domain of the data (e.g. personal data). Moreover, there might be a risk considering the quality of the data used. For example, if a large amount of structurally misannotated data is included in the underlying data set, the central assumption could be affected.

**Main Review:**

The paper considers the problem setting of PML from an information-theoretic perspective and justifies the proposed method MILI-PML accordingly. With this perspective, the authors provide a contribution to the field of PML. The authors include recent methods and compare the performance of the proposed method against those in the experimental part.

The quality of the paper is not sufficient yet and needs improvement. Some details of the theoretical analysis need to be clarified, and the experimental part needs to be extended by additional assessments, especially concerning the limitations of the proposed method.

The paper makes two assumptions on which MILI-PML is based: (1) "that the overall correlation between a ground-truth label $y_k$ and features should be stronger than the overall correlation between a false-positive label $y_{\bar{k}}$ and features" and (2) "It is naturally hold that the KL-divergence value should be smaller when the ground-truth label $y_k$ is added to the existing identified label set $v$ than that of the false-positive label $y_{\bar{k}}$".

The first assumption seems plausible on an intuitive level. However, it is unclear to what extent this assumption is fulfilled in the datasets used. The reported results give little information about the assumption, as only the performance on different evaluation metrics is reported. Whether the assumption tends to be fulfilled within a dataset could be shown by providing measures of the dependency between individual features and labels, such as a suitable correlation coefficient. Further, it is unclear to what extent MILI-PML exploits dependencies between features and labels. Therefore, it would strengthen the paper to assess how the performance of MILI-PML to select or reject a label correctly is related to the level of dependency between the features and a ground-truth or false-positive label. This influence could be analyzed by generating artificial data in which the dependency between a label and the features is controllable and evaluating the performance of MILI-PML under different configurations. This might also indicate potentials limitations of the proposed method.

Regarding the second assumption, I suppose that the relation between $D_{KL}(P(Y_v \cup Y_{\bar{k}} | X) || Q(Y_v \cup Y_{\bar{k}} | X,w))$ and $D_{KL}(P(Y_v \cup Y_{k} | X) || Q(Y_v \cup Y_{k} | X,w))$ depends on both the choice of model $Q$ and its parameterization $w$. I expect that depending on the choice of $Q$ and $w$, the relation of the two KL-divergences can take on any value. Please clarify why your assumption that in general $Q$ approximates the true probability distribution $P$ better when a ground-truth label is added to $v$ rather than a false-positive one holds.

In Chapter 6.2 (and Chapter 3 of the Supplementary Material), it is stated that the hyper-parameters of the baselines were set to the corresponding values of the original papers. Optimize the hyper-parameters for the baselines for a fair comparison with MILI-PML and state the range of hyper-parameters considered and methods used for selection.

The submission is generally well written and organized. Mathematical statements are presented in (great) detail and are mostly easy to follow. Please see section „Questions and minor remarks“ for minor issues. However, I would recommend a language review before publication.

With the provided information about the implementation and the experiments, it is not yet possible to reproduce the results. Therefore, please provide the following details to improve reproducibility:

Number of evaluation runs and calculation of error bars in the results tables.
Detailed descriptions of how the first and third quantities are calculated in Eq. 21.
Detailed descriptions of how the real world datasets are used in the experimental part. For example, the MIRFlickr dataset [1] contains Flickr tags (raw form and processed form) and annotations. It is unclear which of these are used as labels, as the relevant statistics do not match the information on the website.
The experimental section indicates promising results for MILI-PML and could be of significance for the area of PML due to both the performance improvement and the theoretical analysis. However, the experimental part does not yet evaluate the method thoroughly enough, and therefore the significance cannot yet be assessed finally.

Questions and minor remarks:

Can you explain the change of distribution in Eq. 7 in which the joint probability distribution of features and labels $P(x,y)$ is exchanged with the joint probability distribution of disregarded labels and selected labels $P(y_{\bar{v}, y_v})$?

Minus is missing at the last line of Eq. 1 in the supplementary material.

Typos or missing words in line: 56, 79, 136, 147, 151, 175, 184, 258, 278, 283.

[1] http://press.liacs.nl/mirflickr/

**Time Spent Reviewing:**

3

---

> ### Author Response · Authors · 2021-08-09
> **Response**
>
> Thanks for your constructive comments and suggestions. We address your main concerns as follows:
>
> Question: The first assumption seems plausible on an intuitive level. However, it is unclear to what extent this assumption is fulfilled in the datasets used.
>
> Answer:
> To resolve your concern, we employ Pearson Correlation Coefficient (PCC) and Normalized Mutual Information (NMI) as measures to evaluate the dependency between individual features and labels on an artificial dataset (e.g., CAL500), with different configurations (e.g., #CandidateLabes =55, 75). This measurement can show to what extent the first assumption is fulfilled. We use r_{X,Y_v}, I_{X,Y_v} to denote the PCC and NMI between individual features and ground-truth labels, and r_{X,Y_\bar{v}} and I_{X,Y_\bar{v}} to denote the PCC and NMI between individual features and false-positive labels. The false-positive labels are configured randomly by choosing from the remaining labels after removing the ground-truth labels of each instance. The PCC is calculated by the mean value of PCC between each feature vector and each label vector. Under the configuration of #CandidateLabes =55, we have r_{X,Y_v}=0.0035, r_{X,Y_\bar{v}}=0.0013, I_{X,Y_v}=0.0984 and I_{X,Y_\bar{v}}=0.0681, while under the configuration of #CandidateLabes =75, the values are 0.0035, 0.0019, 0.0984, 0.0736 respectively.
>
> From the results, we can see that the correlations between features and ground-truth labels are consistently higher than that of features and false-positive labels in terms of both PCC and NMI, which verifies the correctness of the first assumption.
>
> Interestingly, the correlations are stronger under NMI than that under PCC, which may be because mutual information also considers the nonlinear correlations. We believe these results are able to validate the effectiveness of our work using mutual information to measure the dependency between features and labels to some extent.
>
>
> Question: Further, it is unclear to what extent MILI-PML exploits dependencies between features and labels.
>
> Answer: We further conduct experiments to evaluate the performance of the proposed method in terms of average precision with increasing iterations on an artificial dataset (e.g., CAL500) under the configuration of #CandidateLabes = 65, and we calculate the corresponding Pearson Correlation Coefficient (PCC) and Normalized Mutual Information (NMI) of features and ground-truth labels, which are denoted as r_{X,Y_v} and I_{X,Y_v} respectively. These results are reported at iterations=10, 20, 50, 100 in the following table.
>
> Iterations | r_{X,Y_v} | I_{X,Y_v} | average precision
>
> 10	|  0.0014	| 0.0694 |	18.65%
> 20	|  0.0017	| 0.0721 |	31.27%
> 50	|  0.0026	| 0.0889 |	42.11%
> 100	|  0.0032	| 0.0908 |	58.54%
>
> From the results, we can see that with the increasing iterations, the correlations between features and ground-truth labels are increasing in terms of both PCC and NMI, and the average precision of the proposed method (MILI-PML) is also increasing, which illustrates the performance of MILI-PML is highly correlated with dependencies between features and labels; namely, the dependencies between features and labels are highly exploited in training MILI-PML models.
>
>
> Question: Regarding the second assumption, …; Please clarify why your assumption that in general Q approximates the true probability distribution P better when a ground-truth label is added to v rather than a false-positive one holds.
>
> Answer: The second assumption is reasonable. As indicated in the manuscipt, P(y_v|x) is the true distribution of ground-truth labels given features, while Q is the predictive model expected to approximate the true distribution of ground-truth labels, rather than approximate the distribution of false-positive labels. Thus, Q can approximate the true distribution better when a ground-truth label is added to v rather than a false-positive one.
>
>
> Answer to Other Questions:
>
> In the experiments, we tried a range of hyper-parameters for the baseline methods with five-fold cross validation, and find the baselines perform the best when following the original settings in the corresponding references. We subsequently perform five-fold cross validation on each dataset and report the average results of each method in terms of ranking loss, hamming loss, one error, coverage, and average precision evaluation metrics.
>
> The MIRFlickr dataset is pre-processed following the reference [14] in the manuscript. We would like to clarify this in the revised manuscript.
>
> In Eq. 7, as the logP(y_\bar{v}|y_v) term does not involve x, we can derive the “seems exchange part” based on the relation between the joint distribution and the marginal distribution.
>
> We would like to clarify the configurations of the predictive model Q. As indicated in the experiment (lines 241-243), Q is specified as Binary Relevance (BR), which is implemented by LIBLINEAR with L2-regularized square hinge loss.
>
> We would like to add the “Boader Impact” section and have the language proofread by a professional English editor in the revised version.

---

### Official Review · Reviewer_yXKv · 2021-07-14

**Rating:** 8
**Confidence:** 4

**Summary:**

This paper presents two fundamental questions about the assumption of existing PML research. The authors derive a novel PML method from a probabilistic formulation which meets the $\epsilon$-identifiable definition. They provide theoretical analysis and algorithms to verify the proposed two questions. Thorough comparative experiments validate the superiorities of the proposed method.

**Limitations And Societal Impact:**

1.	From Table 3 to 5, we can see that the proposed method does not always perform the best, such as, onYeastBP and Eurlex-ed datasets. How do the authors view this?
2.	How to calculate the probabilities in Eq. (18) and Eq. (21)?
3.	How is the L2-regularized square hinge loss incorporated into the optimization algorithm designed in section 5?
4.	In Algorithm 2, the while loop of the condition should be explicitly explained.
5.	There are some typos, such as “fundermental”, “ojective”. The authors should carefully proofread this paper and correct all the typos.


**Main Review:**

Pros:
This paper provides a new insight into partial multi-label learning problem from mutual information perspective, which is novel and highly related to the NeurIPS community. The presentation of the paper is clear and the relation with prior work is well-explained. A new objective function for PML is derived from a probabilistic formulation and a novel definition to evaluate whether a PML method can identify the ground-truth labels precisely is presented. The theoretical analysis is sound and solid. Thorough comparative experiments have been conducted to validate the superiorities of the proposed method, which has good academic value.

Cons:
1.	From Table 3 to 5, we can see that the proposed method does not always perform the best, such as, onYeastBP and Eurlex-ed datasets. How do the authors view this?
2.	Some details need elaboration. How to calculate the probabilities in Eq. (18) and Eq. (21)? Besides, it is pointed out that LIBLINEAR with L2-regularized square hinge loss is employed to train the binary classifiers. How is the L2-regularized square hinge loss incorporated into the optimization algorithm designed in section 5?
3.	In Algorithm 2, the while loop of the condition should be explicitly explained.
4.	There are some typos, such as “fundermental”, “ojective”. The authors should carefully proofread this paper and correct all the typos.


**Time Spent Reviewing:**

1.5 hour

---

> ### Author Response · Authors · 2021-08-09
> **Response**
>
> Thanks for your constructive comments and suggestions. We address your main concerns as follows:
>
> Question: From Table 3 to 5, we can see that the proposed method does not always perform the best, such as, onYeastBP and Eurlex-ed datasets. How do the authors view this?
>
> Answer: The proposed method is a little bit inferior but comparable to the best performance on the YeastBP and Eurlex-ed datasets. It might be because the correlations between features and labels are not as strong as that of other datasets, where the information captured by mutual information is not enough in training the MILI-PML models and this increases the difficulty in identifying the ground-truth labels. However, the overall performance of the proposed method is the best on the other datasets.
>
>
> Question: Some details need elaboration.
>
> Answer: L2-regularized square hinge loss for the binary classifiers is incorporated into Step 3 of Algorithm 1. Step 3 aims to solve a multi-label classification problem. We choose BR as the multi-label classifier in Eq. (17) or equivalently Eq.(18), and use SVM (L2-regularized square hinge loss) to train the binary classifiers. Eq. (18) is used to calculate the model parameter w after the ground-truth label set v is fixed. At Step 3 of Algorithm 1, PML problem becomes a multi-label classification (MLC) problem with fixed multi-label dataset. We adopt BR method and use SVM (L2-regularized square hinge loss) in LIBLINEAR to train the binary classifiers for the MLC problem. Eq. (21) is used to calculate the ground-truth label set parameter v. After the model parameter w is calculated by Eq. (18) in Step 3, Eq. (21) can be calculated by choosing label y_k from candidate label set.
>
>
> Question: In Algorithm 2, the while loop of the condition should be explicitly explained.
>
> Answer: The while loop condition is formulated with the conditional mutual information between a candidate label y_k and all features on condition of the already identified ground-truth label set v.
>
>
> Question: There are some typos, such as “fundermental”, “ojective”. The authors should carefully proofread this paper and correct all the typos.
>
> Answer: We would like to have the language proofread by a professional English editor in the revised paper.

---

### Official Review · Reviewer_gNGy · 2021-07-16

**Rating:** 7
**Confidence:** 4

**Summary:**

This paper tackles the partial multi-label learning problem from a probabilistic formulation with theoretical interpretation and derives a novel objective function based on the concept of Mutual Information (MI). To optimize the objective function, the authors propose an optimization algorithm to identify ground-truth labels and update classifier alternatively.

**Limitations And Societal Impact:**

- Why there are no related references following the statement “the state-of-the-art PML methods” in the first section?
- In the paper, the authors claim that y, y_v are the vectors, but describe that “y_v is selected from y”. I believe there are some notation abuses, and the authors should modify it to make the mathematical expression more strict.
- The notation v is used as vector in section 3, but used as the identified ground-truth label set in section 4. Different notations are suggested to use.


**Main Review:**

The idea of studying the PML problem with tools from information theory is novel. The training objective function derived by the authors is reasonable and can be well explained theoretically. The technique and theoretical parts are solid and seem sound to me, which can benefit the following research in PML. The contribution is highly related and important to the community.

The overall structure and writing are clear and easy to follow.

I have some questions:
- Why there are no related references following the statement “the state-of-the-art PML methods” in the first section?
- In the paper, the authors claim that y, y_v are the vectors, but describe that “y_v is selected from y”. I believe there are some notation abuses, and the authors should modify it to make the mathematical expression more strict.
- The notation v is used as vector in section 3, but used as the identified ground-truth label set in section 4. Different notations are suggested to use.


**Time Spent Reviewing:**

2

---

> ### Author Response · Authors · 2021-08-09
> **Response**
>
> Question: Why there are no related references following the statement “the state-of-the-art PML methods” in the first section?
>
> Answer: Thank you for the comment. We would like to add the related references in the revised version.
>
> Question: In the paper, the authors claim that y, y_v are the vectors, but describe that “y_v is selected from y”. I believe there are some notation abuses, and the authors should modify it to make the mathematical expression more strict.
>
> Answer: We are sorry for the confusing. We would like to improve the notations in the revised version.
>
>
> Question: The notation v is used as vector in section 3, but used as the identified ground-truth label set in section 4. Different notations are suggested to use.
>
> Answer: Thank you for your suggestion. We will use different notations to represent vector and label set in the revised version.

---

### Decision · Program_Chairs · 2021-09-27

**Decision:**

Accept (Poster)

**Comment:**

The author response clarified the main concerns regarding the paper, notably regarding the assumptions and the presentation. The reviewers agree that the paper makes a strong contribution to the PML setting, and that the experiments convincingly support the approach.